# Up-Sampling Method for Low-Resolution LiDAR Point Cloud to Enhance 3D Object Detection in an Autonomous Driving Environment

**DOI:** 10.3390/s23010322

**Published:** 2022-12-28

**Authors:** Jihwan You, Young-Keun Kim

**Affiliations:** School of Mechanical and Control Engineering, Handong Global University, Pohang 37554, Republic of Korea

**Keywords:** 3D upsampling, LiDAR super-resolution, interpolation, 3D object detection

## Abstract

Automobile datasets for 3D object detection are typically obtained using expensive high-resolution rotating LiDAR with 64 or more channels (Chs). However, the research budget may be limited such that only a low-resolution LiDAR of 32-Ch or lower can be used. The lower the resolution of the point cloud, the lower the detection accuracy. This study proposes a simple and effective method to up-sample low-resolution point cloud input that enhances the 3D object detection output by reconstructing objects in the sparse point cloud data to produce more dense data. First, the 3D point cloud dataset is converted into a 2D range image with four channels: x, y, z, and intensity. The interpolation on the empty space is calculated based on both the pixel distance and range values of six neighbor points to conserve the shapes of the original object during the reconstruction process. This method solves the over-smoothing problem faced by the conventional interpolation methods, and improves the operational speed and object detection performance when compared to the recent deep-learning-based super-resolution methods. Furthermore, the effectiveness of the up-sampling method on the 3D detection was validated by applying it to baseline 32-Ch point cloud data, which were then selected as the input to a point-pillar detection model. The 3D object detection result on the KITTI dataset demonstrates that the proposed method could increase the mAP (mean average precision) of pedestrians, cyclists, and cars by 9.2%p, 6.3%p, and 5.9%p, respectively, when compared to the baseline of the low-resolution 32-Ch LiDAR input. In future works, various dataset environments apart from autonomous driving will be analyzed.

## 1. Introduction

Light detection and ranging (LiDAR) is a core mapping and detection sensor technology that ensures the safe and autonomous navigation of robots, drones, and vehicles. LiDAR is typically implemented in autonomous driving applications for 3D object detection, and particularly the detection of pedestrians, cyclists, and vehicles from a few to hundreds of meters. 3D LiDAR with high vertical resolution, such as 64-channel (Ch) LiDAR, can achieve highly accurate 3D detection of relatively small objects within a large volume area. Consequently, it is used in autonomous vehicles to obtain a more vertical dense point cloud and to achieve better object segmentation and a safer driving experience.

Furthermore, the sparsity of the point cloud increases with the increase in the distance of the object owing to the fixed vertical angles of the laser diodes of a 3D LiDAR. A low-resolution LiDAR, such as 16-Ch, may not be able to effectively capture the points required to distinguish a pedestrian from the background, causing major safety issues.

Accordingly, the representative open datasets of LiDAR 3D object detection, such as the KITTI [1] and Waymo Open [2] datasets, obtain the data point clouds of the driving environment by using 64-Ch LiDARs.

A high-resolution LiDAR, although preferable for accurate detection and safe driving, is expensive. A 64-Ch LiDAR is approximately 10–20 times more expensive than a low-resolution 16-Ch LiDAR. The high cost of high-resolution LiDAR considerably restricts its widespread implementation in commercial vehicles. This cost can be reduced through the super-resolution of the sparse point cloud of a low-resolution LiDAR while achieving a high 3D object detection accuracy.

The conventional 3D data up-sampling methods include bilinear or bicubic interpolations [3]. Although they present a high processing speed, they overly smoothen the object edges. Most existing deep-learning super-resolution models are designed for 2D images, and only a few models can be implemented for 3D point clouds. PU-Net [4] proposed a method for up-sampling 3D point cloud models; however, they primarily focused on obtaining the resolution of small objects. Shan et al. [5] proposed a deep-learning model to increase the LiDAR resolution, wherein the super-resolution effects of the model on the simulation data were compared with those of other conventional methods. However, object detection enhancement due to the super-resolution was not analyzed and validated. Contrary to such methods, we propose a simple and efficient method for the up-sampling of the point cloud based on a non-deep-learning model. This method enhances the 3D object detection accuracy by predicting and filling the sparse space in the point cloud data obtained by a low-resolution LiDAR. The proposed method also solves the over-smoothing problem of the conventional interpolation methods while achieving higher operational speed and better object detection performance when compared to the aforementioned deep-learning 3D super-resolution method [4,5]. The proposed method first converts the 3D point cloud data into 2D range images using four channels. It then implements a weighted interpolation with a decaying factor and excludes the zero value outliers. Subsequently, the up-sampled point cloud is passed to the widely used point-pillar [6] 3D object detection model to evaluate the improvement in the detection accuracy.

The contributions of this paper can be summarized as follows:This study presents a simple and efficient up-sampling method for the sparse point cloud of a low-resolution LiDAR,The proposed up-sampling method enhances the 3D object detection of a low-resolution LiDAR,A comparative analysis is performed using the conventional interpolation methods and recent deep-learning super-resolution models on point clouds for 3D object detection.

## 2. Related Work

### 2.1. 2D Image Up-Sampling

Previous studies conducted on super-resolution were primarily focused on the enhancement of a single 2D image [7]; in this process, a low-resolution image with coarse details is converted to a corresponding high-resolution image with refined details. Other similar terminologies for super-resolution in the research community include image scaling, interpolation, and up-sampling. In the current study, we use the term up-sampling to refer to super-resolution.

The conventional image up-sampling methods employ interpolation methods that fill an empty space with data points using a low-order polynomial or other general kernels. The most commonly used interpolation approaches on 2D images include the nearest-neighbor, bilinear, bicubic, and lanczos [7] schemes; these methods are simple and present a low computational overhead. However, the visual quality obtained is low due to the heavy smoothing performed on the edges.

Deep-learning methods, which employ highly non-linear kernels, present better up-sampling performance for complex scenarios. The super-resolution convolutional neural network (SR-CNN) [8] is a pioneer end-to-end up-sampling model that uses several convolution layers stacked on top of each other. The efficient sub-pixel convolutional neural network (ESPCN) [9] is a more complex and faster up-sampling method that extracts several features in the low-resolution feature map that are aggregated to reconstruct a high-resolution image output. Additionally, various approaches of deep-learning have been presented for image up-sampling, such as residual networks methods [10,11], recursive network methods [12,13], progressive reconstruction designs [14,15], densely connected networks [16,17], multi-branch designs [18,19], attention-based networks [20,21], multiple degradation handling networks [22,23], and GAN (generative adversarial networks)-based models [24,25].

### 2.2. 3D Point Cloud Up-Sampling

The 3D point cloud up-sampling process requires interpolation in a higher-dimension space, including the depth map, when compared to the 2D image up-sampling process. Typically, the low-resolution point cloud data is a 3D or higher-dimension space comprising an empty space with sparse, unordered data points, which increases the complexity of the interpolation.

#### 2.2.1. 3D Up-Sampling with RGB Image

Most existing studies on depth map up-sampling have combined corresponding 2D RGB images. The existing image-guided up-sampling methods use fusion filters such as Markov random field (MRF) [26,27] bilateral filtering with pixel color information [28]. The depth interpolation is estimated using a filter based on the information of the neighboring pixels such as the geometric distance, intensity, and anisotropic diffusion [29]. Some recent studies have introduced deep-learning models for image-guided 3D up-sampling. Shan et al. [5] projected the 3D point cloud as a 2D range image and up-sampled the cloud to a denser map using an encoder–decoder architecture with residual connections similar to REDNet [11] and UNet [30]. Although the reconstruction loss is lower when compared to conventional interpolation methods, the simulation-based method [5] did not demonstrate the effectiveness of reconstruction in improving 3D object detection accuracy.

#### 2.2.2. 3D Up-Sampling with Point Cloud

For the applications where only LiDAR is available, the 3D up-sampling of the sparse point cloud must be performed without using any other high-resolution guides. Only a few studies have been conducted on LiDAR up-sampling methods that map the sparse 3D points into a higher resolution using deep-learning models.

PU-Net [4] is a recently developed data-driven up-sampling method on the 3D point cloud data; it learns the multi-level features of each point and implicitly expands the point sets via multi-branch convolution units in feature spaces. This model performs the up-sampling of small-sized objects in indoor environments. The reconstruction loss involved is lower when compared to conventional interpolation methods; however, the effectiveness of the reconstruction in enhancing the 3D object detection accuracy was not demonstrated.

The proposed up-sampling method for low-resolution LiDAR point clouds directly enhances the 3D object detection. The proposed method is not based on deep learning, contrary to the recent methods; hence, it achieves more efficient calculations. The reconstructed 3D point clouds are then fed to the 3D object detection model to evaluate the effectiveness of the object detection. Furthermore, we validated the up-sampling effectiveness of the proposed method by performing a comparative analysis of our method with the recently developed deep-learning method [5] and the conventional interpolation methods in terms of the object detection performance on the KITTI dataset.

## 3. Comparison Methods

### 3.1. Test Data Preparation

This section presents the KITTI dataset [1] of the bird’s-eye view used for 3D object detection. The dataset was acquired using the Velodyne HDL 64-Ch LiDAR and a total of 7481 frames were used for training. High-resolution LiDAR range images were obtained by converting the 64-Ch LiDAR point cloud data into range. The low-resolution range images were obtained by extracting some rows out of the 64-Ch LiDAR range images. In the following section, we present a comparison of the up-sampling performance between the low-resolution (16-Ch) LiDAR point cloud and the high-resolution point cloud (64-Ch) of the conventional and proposed methods.

### 3.2. Conventional Up-Sampling

The representative conventional up-sampling methods were selected for the performance comparison with the proposed up-sampling method. The conventional up-sampling methods include the interpolation-based and deep-learning-based methods, as explained in Section 2.

Figure 1 presents the comparisons of the basic representative interpolation methods, which include the nearest-neighborhood, bilinear, and bicubic interpolations. In Figure 1, the nearest-neighborhood interpolation method cannot describe complex shapes such as curves of objects. Several noisy points are generated around the object in the bilinear and bicubic interpolation. Considering the sparsity characteristic of a range image, it is essential to perform interpolation for all of the points apart from the zero value point in a range image.

### 3.3. Deep Learning Based Up-Sampling

Among the deep-learning-based up-sampling methods, we compared the ESPCN [9] network and the method proposed by Shan et al. [5] as a reference. When the ESPCN [9] model was implemented, the distribution of points did not sufficiently describe the shape of an object, as shown in Figure 2. The simulation-based model [5] presented relatively good concurrence with the shape of the 64-Ch LiDAR; however, there were still some noisy points around the object.

## 4. Proposed 3D Up-Sampling Method

Figure 3 presents the outline of the research strategy used for the proposed method. First, the unordered 3D point cloud data are projected onto ordered range map images with multiple channels. The proposed up-sampling method reconstructs the sparse range image input into a dense 3D point cloud. It is then fed to the 3D object detection model as the input data and the detection performance is evaluated on the KITTI dataset.

### 4.1. Projection to Range Image

The number of 3D points obtained through a single scan of a rotating LiDAR is determined by the vertical channel numbers, vertical resolution ($V_{res}$), horizontal FOV, and horizontal angular resolution ($H_{res}$).

The point cloud data is structured as a list of 3D positions and the intensity value of the laser beam points in the order of laser diode scanning, which may vary based on the LiDAR model. The dimension of the point cloud is expressed as *N* by *D*, where *N* represents the total number of beam points and *D* represents the feature numbers of the beam point, such as the 3D Cartesian coordinates and intensity value. Therefore, the data in the point cloud are in an unordered form in terms of the spatial coordinates, contrary to the 2D image pixels.

The complexity of handling such unordered data can be reduced by projecting the point cloud into a range map, which is spatially ordered using the 2D coordinates of the horizontal and vertical scanning angles, as shown in Figure 4. The dimensions of the range map are *H* by *V* by *C*, where *H* denotes the horizontal azimuth (α) angles, *V* denotes the number of LiDAR vertical channels, and *C* denotes the channel of range measurement and intensity. The up-sampling techniques for 2D images can be implemented easily since the range map is an ordered structure.

The projection of the 3D point cloud data to the range map in the azimuth and vertical angles can be processed using the transformation equation in the spherical coordinates, which is expressed as follows:(1)$$\begin{array}{c} R=\sqrt{x^{2}+y^{2}+z^{2}}\\ \alpha =\arctan(y,x)\\ \omega =\arctan\left(z,\sqrt{x^{2}+y^{2}}\right) \end{array}$$

The dimension of the 3D point cloud from Velodyne LiDAR (HDL-64E) is 2048×64×4 Ch (X-Ch, Y-Ch, Z-Ch, Intensity-Ch), and it can be converted to a range map with the dimension of 2048×64×2 Ch (α-Ch, ω-Ch).

### 4.2. Proposed Up-Sampling Technique

The existing up-sampling methods do not adequately represent the shape of high-resolution LiDAR points when there are meaningless zero values within the pixels of the range image or when estimating the boundary of an object. In this study, we present a novel interpolation technique that minimizes the influence of the outlier points. The robustness of the proposed interpolation method is improved by implementing the following strategies.

#### 4.2.1. Pixel-Distance Weighted Interpolation

The up-sampling is applied in the direction of elevation to increase the density of the vertical angle of the LiDAR channel. The proposed method interpolates the empty spaces of the LiDAR range image using the six surrounding neighbor pixels with weights based on their relative distances from the anchor point, similar to the four-point depth interpolation method proposed by Lim et al. [31]. The six neighborhood pixels for the interpolations are labeled from $P_{1}$ to $P_{6}$, starting from the top-left pixel, as shown in Figure 5. The interpolated value, $P^{\prime }$, is obtained from the weighted sum of the neighbors with the decaying factor against the distance as follows:(2)$$W_{i}=e^{-0.5d_{i}}$$
(3)$$P^{\prime }=\frac{\sum_{i=1}^{6}W_{i}P_{i}}{\sum_{i=1}^{6}W_{i}},$$
where $P_{i}$, $W_{i}$, and $d_{i}$ represent the range values, the weight ratio, and the pixel distance between $P_{i}$ and $P^{\prime }$, respectively.

The interpolated output in the 2D range image is converted to a 3D point cloud to visualize and analyze the interpolation effect, as shown in Figure 6.

The distance-weighted interpolation result presented in Figure 6c depicts a denser 3D output with amplified noisy data owing to the fact that the interpolation with the neighboring points that have zero or maximum range values can be considered as the outlier when compared to the surrounding data. The negative effect of the zero-valued pixels in the interpolation process can be reduced by removing the outlier neighbor pixels. The zero or maximum-valued pixel in the 2D range image refers to the LiDAR beam point that does not project on the object in the measurable range.

#### 4.2.2. Pixel-Distance and Range Weighted Interpolation

These outlier points can be skipped in the weighted sum process using the coefficient, $s_{i}$, which presents a value of 0 for outliers; otherwise, it presents a value of 1.

Furthermore, the up-sampling can be enhanced by readjusting each weight, $W_{i}$, based on the relative depth range among the neighbor pixels. The closer the depth of the pixel is, the more weight is given in the interpolation, which is similar to assuming that the closer range neighbor pixels are likely to be projected on the same object. Then, the modified interpolation equation can be expressed as:(4)$$P^{\prime }=\frac{\sum_{i=1}^{6}s_{i}W_{i}P_{i}}{\sum_{i=1}^{6}s_{i}W_{i}}$$
with:(5)$$W_{i}=e^{-0.5d_{i}}\cdot \left(\frac{2}{1+e^{\left(R_{i}-R_{\min}\right)}}\right),$$
where $R_{i}$ is the range value of each neighbor data and $R_{\min}$ is the minimum range value among them. This interpolation result, as shown in Figure 6d, clearly removes the noisy interpolated data for up-sampling enhancement.

#### 4.2.3. Increasing Channel Dimension of 2D Range-Image

The interpolation performance can be further improved by increasing the dimensions of the 2D range image. The 1-Ch 2D range image can be converted to a higher-dimension 2D range image with 4-Ch of the Cartesian coordinates (*x*, *y*, *z*) and intensity values. The values of *x*, *y*, and *z* can easily be obtained from the azimuth, elevation, and range values as:(6)$$\begin{array}{c} x=r\cos\omega \cos\alpha \\ y=r\cos\omega \sin\alpha \\ z=r\sin\omega , \end{array}$$
where azimuth (α) and elevation (ω) represent the coordinates of the 2D range image, and *r* denotes the range value of the pixel.

Figure 6 presents the up-sampling result of the proposed method on a pedestrian dataset. The result demonstrates that this method closely resembles the 64-Ch ground truth. Additionally, the proposed interpolation method presents the highest resemblance to the reference interpolated output when compared to the conventional methods presented in Figure 1 and Figure 2.

## 5. Up-Sampling Effect on 3D Object Detection

### 5.1. Experiment Overview

The effectiveness of the proposed LiDAR up-sampling method was demonstrated by feeding it as the input to the selected 3D object detection model and evaluating the detection accuracy. The point-pillar model [6], which is an effective and widely used 3D detection model with LiDAR data, was used as the 3D model. This model was trained using the dataset up-sampled via the proposed method from the baseline point cloud data.

The baseline dataset comprises the point cloud data of a low-resolution (32-Ch) LiDAR, which was generated by down-sampling the KITTI [1] 3D dataset of a 64-Ch LiDAR.

The numbers of data points in the training, validation, and test datasets were 7481, 1856, and 3769, respectively. The overall performances of the 3D detection task were evaluated based on the mAP (mean average precision) of easy, moderate, and hard difficulty for each class of cars, pedestrians, and cyclists, which are the most common obstacles in an autonomous driving environment.

### 5.2. Algorithm Design (Ablation Studies)

An ablation study was conducted to analyze the effect of the following methods: (A) pixel-distance weighted interpolation, (B) pixel and range weighted interpolation, (C) additional channel on the 2D range image, and (D) ground removal on the 3D object detection. The baseline dataset of the 32-Ch point cloud was up-sampled to a 64-Ch point cloud using the aforementioned methods and the accuracy of the 3D object detection was evaluated for comparison.

#### 5.2.1. (A) Effect of Pixel Distance Weight Interpolation

The baseline dataset was up-sampled to a 64-Ch point cloud using only the six-point pixel distance weighted interpolation based on Equations (Equation 2) and (Equation 3).

This method presented inferior detection results for all of the classes when compared to the detection output with the baseline dataset of the 32-Ch point cloud, as described in Table 1. The interpolation performed using only the pixel distances of the neighbor points, similar to the method reported in [31], raised the 3D detection accuracy from 31.32% to 43.85%.

#### 5.2.2. (B) Effect of Pixel Distance and Range Weight Interpolation

The up-sampling method was enhanced by neglecting the zero-value range points to avoid the outlier effect on the interpolation. Additionally, the range difference between the anchor point and neighbors was considered to apply a higher weight on the points that had a higher chance to be on the same object surface. The data obtained from the interpolation performed using Equations (Equation 4) and (Equation 5) were applied to the baseline dataset and used as the training data for the 3D detection model.

The mAP of the overall classes for all of the difficulty levels increased by up to 3.6% when compared to the baseline result, as presented in Table 1. The mAP of the car detection task increased to 72.39%, when compared to the mAP of 65.82% achieved by the baseline. However, the mAP of the cyclist detection task decreased slightly.

#### 5.2.3. (C) Effect on Increasing Range Image Channels

In the previous sections, the point clouds were up-sampled to a 2D range image using 2-Ch of the range and intensity values. The axes of the image are the azimuth and vertical angles, as shown in Figure 4.

The 2D image with a higher number of channels was generated using the Cartesian 3D coordinates of each point with Equation (Equation 1). Therefore, the input image used for the interpolation was a 2D image with 4-Ch of the x, y, and z-axis position values and the intensity value.

The detection achieved by the up-sampled image of the 4-Ch presented superior performance when compared to the baseline. The overall mAP for all the classes was enhanced by as much as 8%. The mAP of the easy level cyclist detection increased from 58.1% of the baseline to 64.95%. The overall mAP was enhanced by 5.01%p, 4.59%p, and 4.47%p for the easy, moderate, and hard levels, respectively, when compared to the proposed interpolation method with the 2-Ch image.

Figure 7 and Figure 8 show the 3D object detection results of the selected scene for pedestrians and vehicles, respectively. The up-sampled point cloud and the detection results closely resemble the ground truth. This helped in accurately reconstructing the objects for locating the pedestrians and vehicles.

#### 5.2.4. (D) Effect on Ground-Removal Pre-Processing

A high proportion of the point cloud data included the points on the ground. The effect on the 3D detection accuracy due to the ground-removal pre-processing performed before the up-sampling was analyzed.

The ground removal process was simplified by discarding all of the points located at a lower elevation than the ego-vehicle. Although this approach can work effectively in most driving scenarios, it may fail if the relative angle between the ground and the ego-vehicle is not flat. In this study, we applied the ground removal algorithm presented by [32], which is based on the inclination angle of LiDAR beams corresponding to the horizontal axis. The up-sampling of the 4-Ch image was conducted after the ground removal processing.

The 3D detection results presented in Table 1 demonstrated that the ground removal decreased the detection accuracy, indicating that the ground points have important roles in the detection model that could not be ignored.

### 5.3. Comparison with Previous Methods

The performance enhancement of the 3D object detection task owing to the use of the proposed up-sampling method was compared with the widely used previous methods and the SOTA (state-of-the-art) method for 3D up-sampling. The same baseline datasets were used for the up-sampling process, whereas the point-pillar model was used for the detection.

Table 2 presents the values of the detection mAP for the classes of pedestrians, cyclists, and cars that were compared to analyze the detection enhancement presented by different up-sampling methods. The strict intersection over union (IoU) of 0.7 was applied for the evaluation. The abbreviations E,M, and *H* represent the difficulty levels of easy, moderate, and hard, respectively.

The CNN-based up-sampling methods of ESPCN [9] and the method proposed by Shan et al. [5] exhibited inferior 3D detection performances. These methods were able to reconstruct a denser point cloud with a low overall loss score. However, they could not accurately reconstruct the object shapes, thereby leading to inferior detection performance.

Table 2 presents a comparison between the moderate-level class detections. It can be clearly observed that the proposed solution (Case C) exhibited the best performance in each class detection when compared to the existing up-sampling methods. The mAP of the overall class (level M) was 45.4%, which is approximately 7% higher than that of the baseline dataset.

## 6. Conclusions

This study proposed an up-sampling method for a low-resolution LiDAR to enhance 3D object detection by reconstructing the objects in sparse point cloud data into data with a higher vertical angle resolution.

The existing 3D up-sampling methods based on deep neural networks focus on decreasing the overall loss in the up-sampling reconstruction process instead of enhancing the object detection performance. Our proposed sampling method is designed as a pre-processing stage to implement an enhanced 3D object detection model. First, we converted the 3D point cloud dataset into a 2D range image with 4-Ch. The interpolation on the empty space was calculated based on both the pixel-distance and range values of six neighbor points to conserve the shapes of the original object during the reconstruction. This approach solved the over-smoothing problem faced by the conventional interpolation methods while presenting higher operational speed and better object detection performance when compared to the aforementioned deep-learning super-resolution methods.

The effectiveness of our proposed up-sampling method on the 3D detection was demonstrated by applying it to baseline 32-Ch point cloud data and then feeding them as the input to a point-pillar detection model. The 3D object detection result on the KITTI dataset demonstrates that the proposed method could increase the mAP (strict IoU with 0.7) of pedestrians, cyclists, and cars by 9.2%p, 6.3%p, and 5.9%p, respectively, when compared to the baseline with a low-resolution 32-Ch LiDAR input.

The scope of this study was limited to the reconstruction of 32-ch LiDAR to 64-ch LiDAR on the KITTI dataset. In future works, various dataset environments apart from autonomous driving will be analyzed. Additionally, further analysis is required to integrate the proposed up-sampling method with the 3D object detection model based on 2D range images such as the range-sparse network (RSN) [33], for the construction of an end-to-end network.

## Figures and Tables

**Figure 1 sensors-23-00322-f001:**
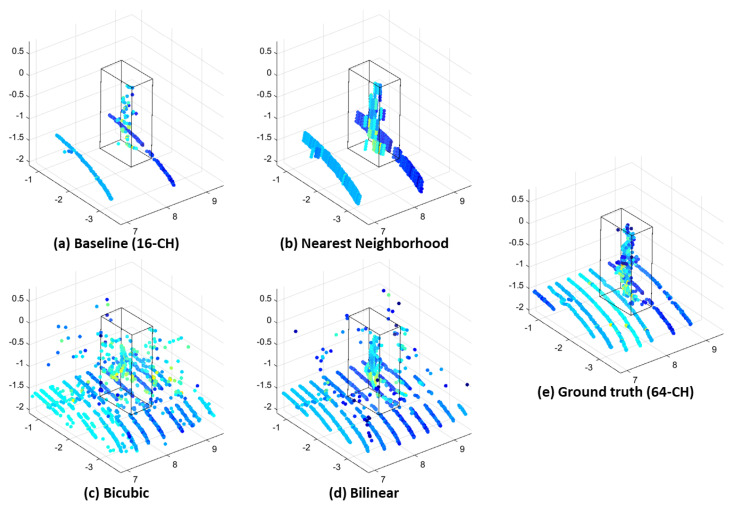
Up−sampled point cloud from 16-Ch (**a**) to 64-Ch (**e**) using classical interpolations (pedestrian). Comparison of (**b**) nearest-neighborhood, (**c**) bilinear, and (**d**) bicubic interpolations.

**Figure 2 sensors-23-00322-f002:**
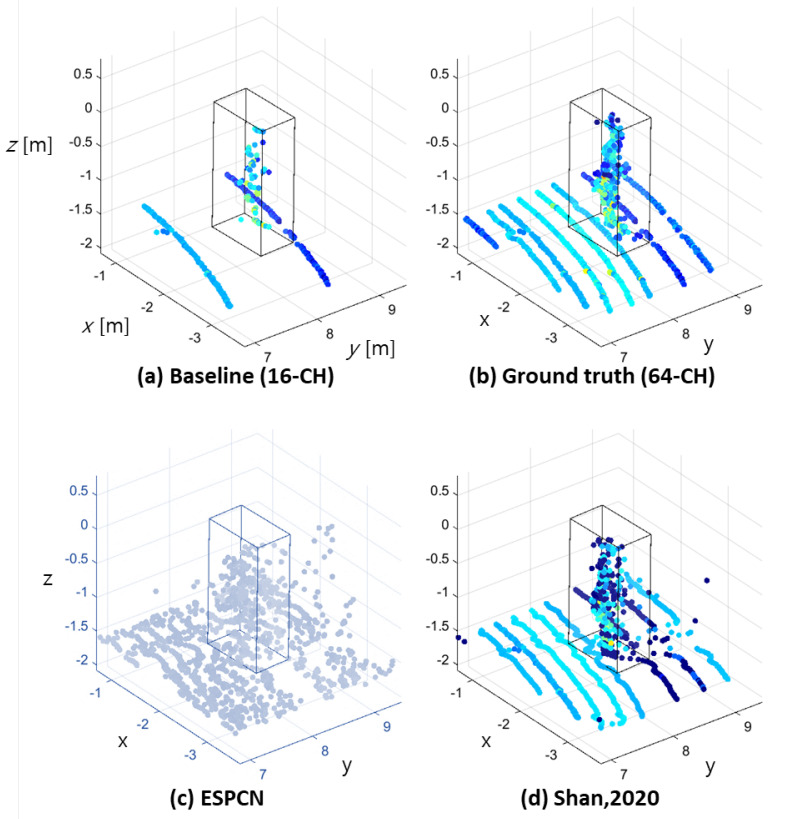
Up−sampled point cloud using deep-learning-based methods (pedestrian). The increase of the vertical angle resolution increases the density of the point cloud along all of the Cartesian axis directions.

**Figure 3 sensors-23-00322-f003:**
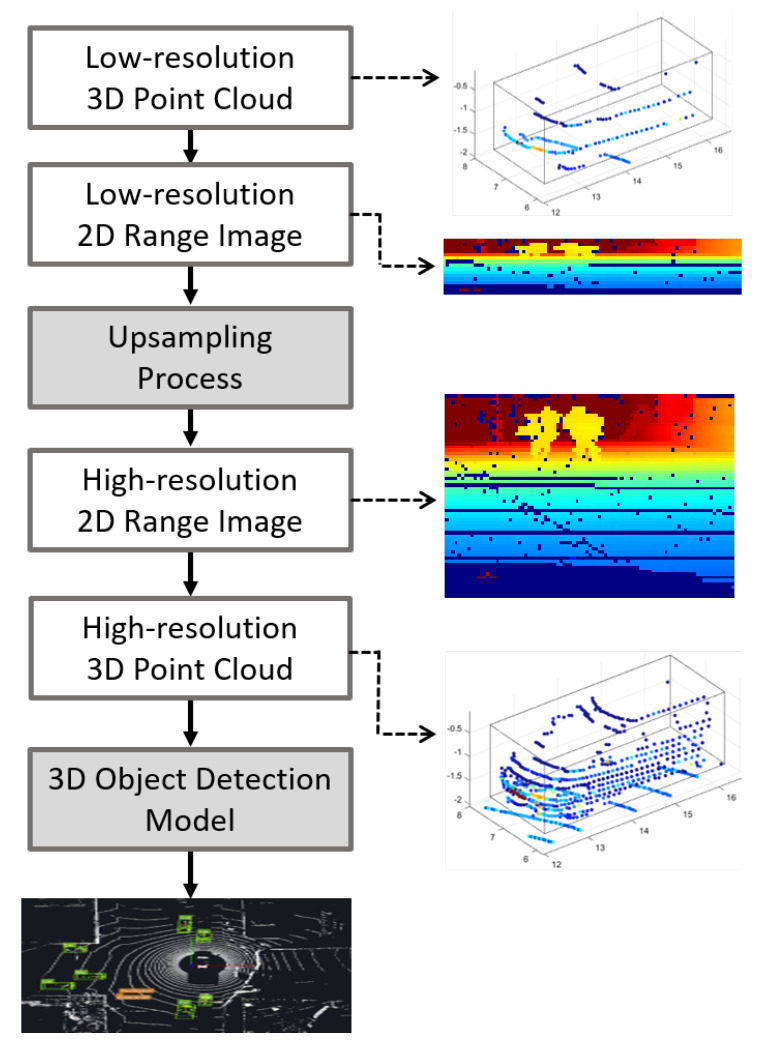
Research strategy for the proposed up-sampling method for 3D detection.

**Figure 4 sensors-23-00322-f004:**
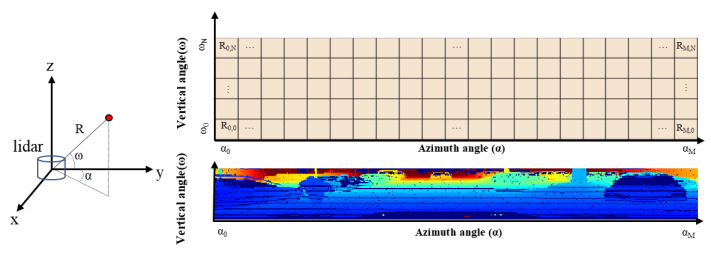
2D Range image from 3D point cloud.

**Figure 5 sensors-23-00322-f005:**
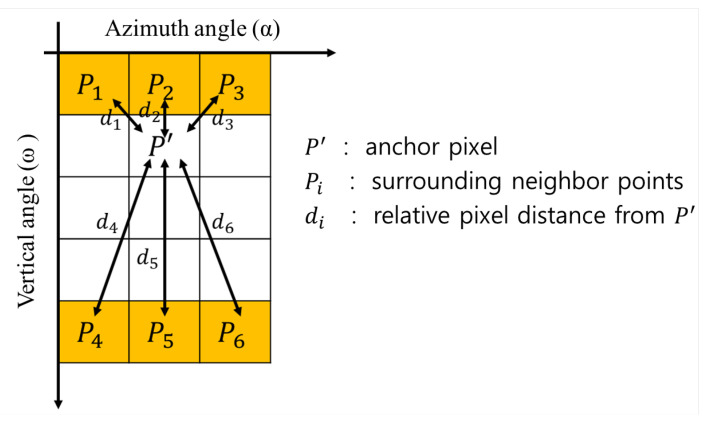
Interpolation with the weighted sum of neighbor pixels. The blank grid represents empty space where there are no LiDAR points.

**Figure 6 sensors-23-00322-f006:**
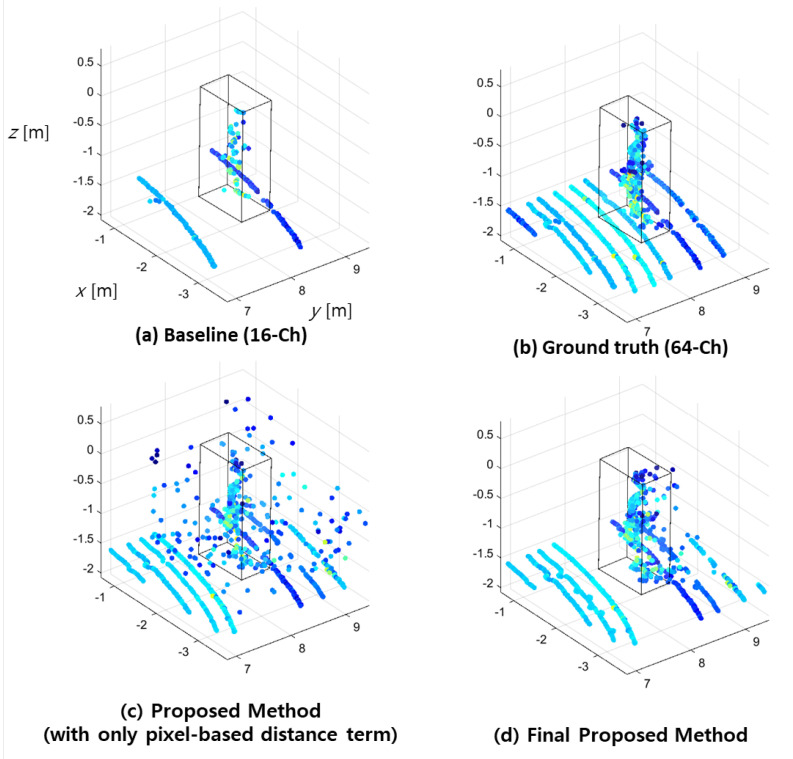
Up-sampled point cloud using proposed methods (pedestrian). Comparison of: (**a**) raw data of 16-Ch LiDAR; (**b**) ground truth 64-Ch LiDAR; (**c**) our proposed method with only pixel-based distance terms; and (**d**) the final proposed method.

**Figure 7 sensors-23-00322-f007:**
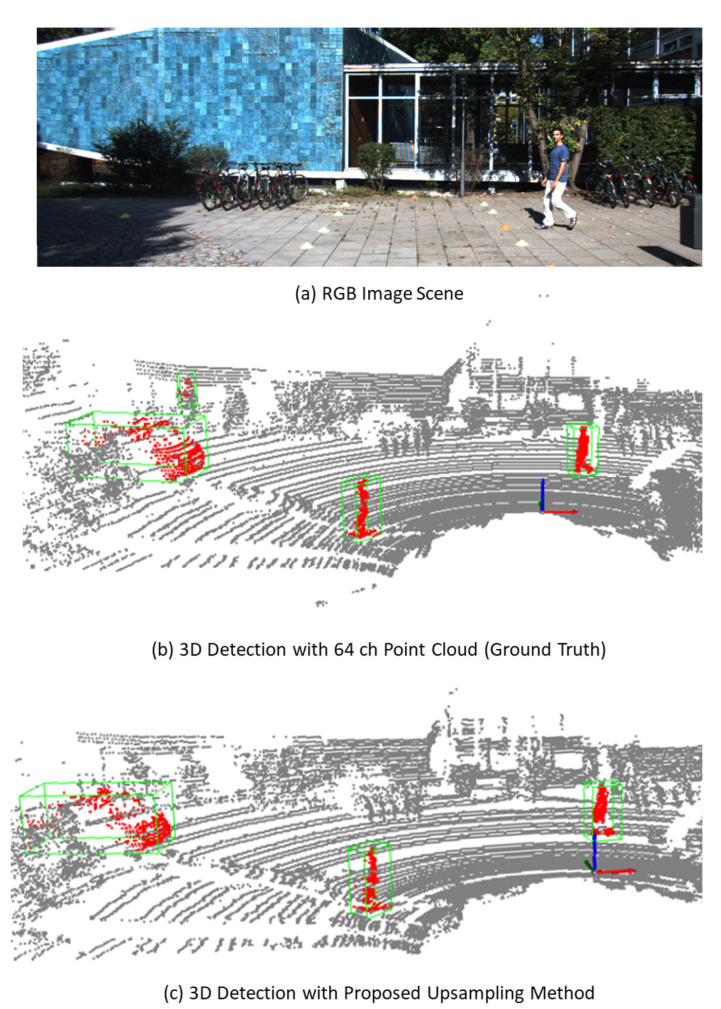
An example of 3D object detection performed using the proposed up-sampling method for pedestrian detection.

**Figure 8 sensors-23-00322-f008:**
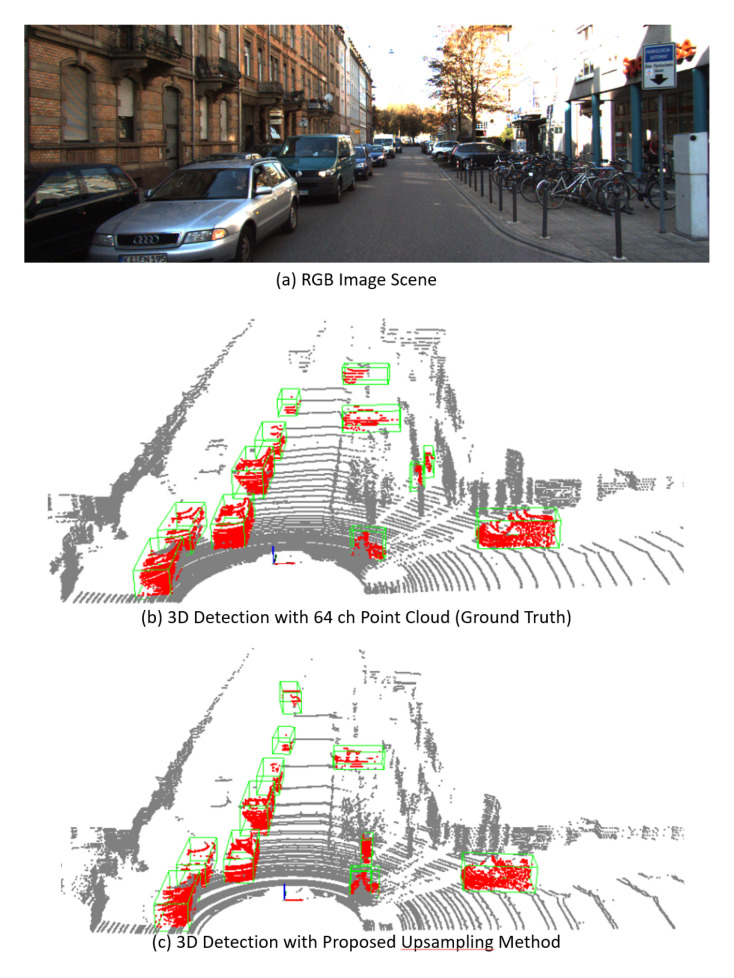
An example of 3D object detection performed using the proposed up-sampling pre-processing for vehicles detection.

**Table 1 sensors-23-00322-t001:** 3D object detection performance (mAP) of the proposed methods on the test dataset.

Class	Baseline (32-Ch)			(A) Pixel Weighted			(B) Pixel and Depth Weighted		
	Easy	Mod.	Hard	Easy	Mod.	Hard	Easy	Mod.	Hard
Pedestrian	30.01	25.98	23.95	27.59	24.94	21.86	36.21	32.31	29.61
Cyclist	58.13	36.88	35.58	52.18	33.02	31.55	56.31	36.59	35.00
Car	65.82	51.88	46.25	51.78	38.15	33.44	72.39	53.41	47.84
Overall	51.32	38.25	35.26	43.85	32.04	28.95	54.97	40.77	37.48
**Class**	**(C) Ground-Removal**			**(D) 2D Range Image of 4-Ch**			**Ground-Truth (64-Ch)**		
	**Easy**	**Mod.**	**Hard**	**Easy**	**Mod.**	**Hard**	**Easy**	**Mod.**	**Hard**
Pedestrian	35.38	31.27	28.31	39.96	35.18	31.71	52.03	46.4	42.48
Cyclist	62.45	40.92	38.19	64.95	43.19	39.95	78.72	59.95	57.25
Car	70.18	55.13	51.60	75.06	57.72	54.19	85.41	73.98	67.76
Overall	56.00	42.44	39.36	59.99	45.36	41.95	72.05	60.11	55.83

**Table 2 sensors-23-00322-t002:** Comparison table of object detection performance using pretrained point-pillars model (up-sampled from 32-Ch to 64-Ch). The strict IoU of 0.7 was applied.

		Ped	Cyc	Car	Overall
Baseline (32-Ch)	E	30.01	58.13	65.82	51.32
	M	25.98	36.88	51.88	38.25
	H	23.95	35.58	46.25	35.26
NN Interpolation	E	27.39	49.09	40.60	39.03
	M	24.22	30.95	30.13	28.43
	H	22.00	29.75	26.99	26.25
Bilinear Interpolation	E	20.67	38.31	30.97	29.98
	M	18.13	24.51	21.12	21.25
	H	16.99	22.95	17.72	19.22
ESPCN [9]	E	2.45	2.90	6.67	4.01
	M	2.57	2.55	5.77	3.63
	H	2.61	2.56	4.55	3.24
Shan,2020 [5]	E	9.29	5.72	5.20	6.74
	M	9.09	9.34	4.55	7.66
	H	9.09	9.29	4.55	7.64
Proposed (Case C)	E	**39.96**	**64.95**	**75.06**	**59.99**
	M	**35.18**	**43.19**	**57.72**	**45.36**
	H	**31.71**	**39.95**	**54.19**	**41.95**

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
