# Peer review of "Up-Sampling Method for Low-Resolution LiDAR Point Cloud to Enhance 3D Object Detection in an Autonomous Driving Environment"

_sensors, 2022, doi:10.3390/s23010322_

Round 1

Reviewer 1 Report

This manuscript describes a simple up-sampling method of low-resolution LiDAR point cloud data to enhance 3D object detection. For this purpose, the 3D point cloud dataset is converted into a 2D range image with four channels (x, y, z, and intensity), and the pixel distance and range values of six neighbor points are employed for the interpolation to avoid the over-smoothing problem. The effectiveness of the proposed method is tested by applying it to baseline 32-Ch point cloud data. Although the manuscript can provide some insight into the preprocessing of LiDAR data in the autonomous driving situation, the authors should reconsider the following issues to improve the presentation quality.

 (major)

L23 “high vertical resolution”, “higher vertical resolution” – the same clause is repeated. The structure of the sentence should be reconsidered to avoid such a repetition. Furthermore, it would be better to mention here that the LiDAR data are assumed to be from a rotating instrument, and the sparseness of the data is associated only with the vertical scan, as clearly stated in Shan et al. [4].

L97 “it” is somewhat ambiguous. Probably, it would be better to use an appropriate term (such as the “simulation-based method”) to refer to the 3D up-sampling method proposed by Shan et al. [4].

L105-110 This part seems like a repetition of L94-98. Please streamline these two subsections by removing the redundancy.

L120 “In this section, the KITTI dataset [1] is used for the experiment.”: Just after this first sentence, please add a brief explanation of the basic features of LiDAR point cloud data available from this dataset. Otherwise, the reader cannot recognize how dense or sparse the data points are (in relation to Fig. 6, for instance) and how many datasets have been analyzed to obtain the results shown in Tables 1 and 2. 

L130-131 First, please explain the overall feature of Fig. 1 in a sentence, such that “Figure 1 shows the results of ….” Also, please explain five panels (a)-(e) in the caption of Fig. 1. Such explanations of panels applies to other figures with multiple panels (Figs. 2, 4, 5, 7, and 8). Please enlarge the fonts in Fig. 1 to warrant readability. The repeated headline “Pedestrian” can be omitted since the information is given in the caption.

L147 “a rotating LiDAR” – this information appears for the first time. This is an essential aspect that should be mentioned in the Abstract and Introduction beforehand.

L163 No indent should be applied to L163. “For the LiDAR model of HDL-64E, the 3D point cloud of dimension 2048 x 64 x 4(x,y,z,intensity) ...” – “For the LiDAR model of HDL-64E (Velodyne), the 3D point cloud of dimension 2048 x 64 x 4 (x*y, z, intensity) ...” If possible, it would be better to specify the values of x and y, not their product of 2048. Please explain the meaning of “4” in “2048 x 64 x 4.”

L164 “a range map of dimension 2048 x 64 x 2.” – “a range map of dimension 2048 x 64 x 2 (\alpha, \omega, intensity).” Please specify the variables and explain the meaning of “2”.

L172 “in the direction of elevation, the y-axis direction on the range image.” – the term “y-axis” is confusing because of the geometrical definitions shown in Fig. 4. Please pay more careful attention to the name of axes before and after the transformation to avoid any misunderstanding. One possible way would be the use of (x, y) and (X, Y), or (\xi, \eta) for indicating different coordinate systems.

L177 Fig. 6: Please explain why the column of P4 – P6 is two columns apart from the anchor pixel. Will these two blank columns be interpolated later? In Fig. 6, would it be better to show the y-axis (elevation) vertically, not horizontally?

L177, L180 The order of Figs. 5 and 6 should be exchanged so that Fig. 5 is described beforehand in the text.

L178 Please explain how the distance d_i is measured (scaled) so that the term 0.5d_i in Eq. (2) becomes dimensionless.

L180 Fig. 5 (current numbering): Please attach panel labels (a) – (d) with an explanation of each panel in the figure caption. Comparison between (a) (16 Ch) and (b) (64 Ch) gives an impression that the interpolation procedure has been applied to the azimuthal coordinate, not the vertical coordinate. Please show the corresponding variable for each axis, and explain why “making the vertical LiDAR channel denser” (L173) results in the change from (a) (input) to (b)-(d) (results).

L209 Please explicitly refer to panels (b) (64 Ch) and (d) of Fig. 5. Please add the axis names in each graph in Fig. 5.

L221-222 “on the mAP of easy, moderate, and hard difficulty classes of cars, pedestrians, and cyclists, respectively, ...” – “on the mAP of three classes with different difficulty levels (easy, moderate, and hard) corresponding to cars, pedestrians, and cyclists, respectively, ...”: what were the percentage fractions of images that included cars, pedestrians, and cyclists among the total of around 13,000 datasets? This information is needed for the exact interpretation of the results shown in Table 1.

L225-227 “the effect of each of the proposed methods: pixel-distance weighted interpolation, pixel and range weighted interpolation, and additional channel on 2D range image and background removal on the 3d object detection.” – “the effect of each of the proposed methods: (A) pixel-distance weighted interpolation, (B) pixel and range weighted interpolation, (C) additional channels on the 2D range image, and (D) background (or ground) removal on the 3D object detection.” This way, the three methods are clearly defined. The symbols (A)-(D) can be employed for the subsection titles (5.2.1 – 5.2.4) as well as the column headers in Table 1.

L277-278 “The meaning of strict and loose in the table column headlines refer to the intersection over union(IoU) of 0.7 and 0.5, respectively.” – no column with the header “strict” and “loose” is found in Table 2. In Table 2, the method “proposed” seems to be the case (C) mentioned above. This point should be explicitly stated.

L286 Please add a paragraph that explains the essential features understood from Figs. 7 and 8. It should be mentioned that the LiDAR images are concentric because of the rotating mechanism assumed in the present analysis.

 (minor)

L0 Please add the university name to the authors’ affiliation.

L2 “channels.” – “channels (Chs).”

L3-4 “the lower is the detection accuracy.” – “the lower the detection accuracy.”

L7 “four channels.” – “four channels (x, y, z, and intensity).”

L8 “values of six neighbor points, to conserve …” – “values of six neighbor points to conserve …”

L13 “KITTI dataset” – “the KITTI dataset”: the same applies to L117.

L15 “mAP” should be spelled out: e.g., “mean AP (average precision).” More strictly speaking, the IoU (intersection over union) level should be specified when referring to mAP.

L18-19 “for safe and autonomous navigations of …” – “for the safe and autonomous navigation of …”

L20 “particularly, detection of pedestrians, …” – “particularly the detection of pedestrians, …”

L23 “channels” – “channels (Chs)”: the abbreviations should be introduced here in the text part (except Abstract). Please note that a consistent use of an acronym should be employed throughout the manuscript, not the arbitrary mixture of “Ch”, “ch,” and “CH” (as found, for example, in Table 1, Table 2, L233, L299, and L303). Also, please note that once the abbreviation “Ch” is introduced, all “64-channel” should be “64-Ch,” as seen on L30, L33, L34, L121, L122, etc.

L23 “such 64-channel LiDAR” – “such as a 64-Ch LiDAR”

L40-41 Please cite a reference (references) for the shortcomings of bilinear or bicubic interpolation.

L43 “PU-Net[3].” – “PU-Net [3].” Please note that a space should be inserted before a parenthesis “[“ or “(“. This rule should be applied to L74, L75, L86, L91, L137, L148, L163, L201, L205, L215, L218, Table 1 column header, L306, etc.

L44 “… was proposed in [4].” – “… was proposed by Shan et al. [4].” The same applies to L94, L136, L138, L175, L264, etc.

L45 “with that of …” – “with those of …”

L52 “a decaying factor” is stated here, but this term never appears hereafter.

L59 The acronym “SOTA” should be explained somewhere in the Introduction.

L63 “mainly focus on …” – “have mainly focused on …”

L86-87 “with sparse unordered data points” – “with sparse, unordered data points”

L92-93 “such as geometric distance, intensity, and anisotropic diffusion he2015sparse.” – “such as the geometric distance, intensity, and anisotropic diffusion.”

L123-124 “16ch,32ch” – “16 and 32 Ch”, “64ch” – “64 Ch”

L151, 152, 157 Please use italic fonts for describing variables such as N, D, H, V, and C.

L157 In Fig. 4, please check if the angles \alpha and \omega are correctly specified in and from the xy plane.

L158 Please use a consistent use of the variable H_res (“res” should be a subscript.)

L158-159 “C is typically range measurement and intensity.” – “C is typically the values of target range and intensity.”

L162 “using the transformation equation in spherical coordinates, as shown in Eq. (1).” – “using the transformation equation in spherical coordinates given as”. Please note that each equation should be part of a sentence. In this regard, it is better to put a period at the end of an equation (or an equation set).

L165 “4.2. Proposed Up-sampling technique” – “4.2. Proposed up-sampling technique,” or “4.2. Proposed Up-sampling Technique.” The same applies to all similar cases. For the subsection header style, please refer to a recent issue of the journal.

L184 Subtitle 4.2.2 “pixel-distance” – “Pixel-distance”

L191 “The closer depth the pixel is, ...” – “The closer depth of the pixel, ...”

L196 No indent is required. “This interpolation result, as shown in Fig. 5,(d) clearly removes ...” – “Figure 5(d) shows the result of this interpolation, with clear removal of ...”

L200-203 “The 2D range image of 1-channel with the values of the range can be converted to a higher  channel dimension of a 2D range image with 4-channels for x-position(x),y-position(y), z-position(z) and intensity values. The values of Cartesian coordinates of x, y, and z can be easily obtained ...” – “The 1-Ch 2D range image with the values of the range can be converted to a higher-dimension, 2D range image with 4-Chs of the Cartesian coordinates (x, y, z) and intensity values. The values of x, y, and z can easily be obtained ...”

L205 The names of the angular coordinates should be consistent with those in Fig. 4.

L207-208 “The progress of ... the reference output.” – This part should be deleted.

L220 “The number of train, validation, and test datasets were ... “ – “The number of training, validation, and test datasets were ...”

L221 The acronym “mAP” should be spelled out with an appropriate reference. Also, the caption of Table 1 should include the information that the quoted values are mAP for each analysis.

L225 “AN ablation study was ...” – “An ablation study was ...”

L231-232 “up-sampled to 64-ch of point cloud” – “up-sampled to 64-Ch point cloud”, “Eq. (2) and Eq. (3)” – “Eqs. (2) and (3).”

L241-242 “The interpolation using Eq. (4) and Eq. (5) were applied to the baseline dataset and used as the train data for the 3D detection model.” – “The interpolation using Eqs. (4) and (5) was applied to the baseline dataset and used as the training data for the 3D detection model.”

L264 “proposed in [30]” – “proposed by Bogoslavski et al. [30]”

L267 “The 3D detection result in Tab. 1 showed that ...” – “The 3D detection result in Table 1 shows that ...”: please use the present tense for referring to figures and tables.

L275 “Table 2 shows the detection mAP for the classes of pedestrians, cyclists, and cars were compared ...” – “Table 2 shows the values of detection mAP for the classes of pedestrians, cyclists, and cars, which are compared ...”

L280-281 “known to reconstruct to a denser point cloud with ...” – “known to reconstruct a denser point cloud with ...”

L283-284 “were compared as in Table 2, it clearly showed the proposed solution exhibited the highest performance ...” – “are compared as in Table 2, it is clearly seen that the proposed solution (C: additional Chs on 2D range image) exhibits the highest performance ...”

L289 Please rephrase “a higher dense data.”

L301 Please describe the IoU level for the present mAP analysis.

L306 “RSN(Range-Sparse Network)[31]” – “RSN (Range-Sparse Network) [31]”

Author Response

We have attached the response letter to reply to the reviewer comments

Reviewer 2 Report

See enclosed file

Author Response

(The authors gave the same response as above.)

Round 2

Reviewer 1 Report

The authors have made significant improvements to the manuscript, which has become mostly understandable. Please consider the following issues for preparing the final version.

 L15 “mAP(mean average precision)” – “mAP (mean average precision)”: Please note that a space is needed between a word and a parenthesis. This is according to the general rule in English that a space is needed to separate two consecutive words. The same applies to many similar cases, including L74, L75, L86, L90, L149, L164-166, L269, L274, etc.

 L24 “high vertical resolution, such 64-channel(Ch) LiDAR, for acquiring a more vertical dense point cloud” – “high vertical resolution, such as a 64-channel (Ch) LiDAR, for acquiring a more vertically dense point cloud”

 L33-34 “64-channel, 16-channel” – “64-Ch, 16-Ch.” Such use of abbreviation should be applied to L52, L118-119, L121, L139, L143, L148, L159, ..., L222, L246-247, L250, L262, L291, L297, L301, etc.

 L35 “the wide implementation of commercial vehicles.” – “the wide implementation on commercial vehicles,” or “the wide implementation of the sensor on commercial vehicles.”

 L47 “In contrast to recent works, ...” – “In contrast to such recent works, ...”

 L50-51 “while performing faster and better object detection results.” – “while performing faster and better object detection,” or “while attaining faster and better object detection performance.” The same applies to L294-295.

 L56-60 Please add two commas, “and”, and a period, as “\dot AAA, \dot BBB, and \dot CCC.”

 L58 The term “pre-processing” appears for the first time in this text. Please add a brief explanation of the standpoint of this up-sampling method in the object detection data processing as a whole (somewhere in the earlier part of the Introduction section.) Otherwise, this terminology should be rephrased.

 L81-82 “attention based, GAN based” – “attention-based, GAN-based”: it would be better to spell out “GAN” as “GAN (...).”

 L104 “multi-level features per point” – “multi-level features of each point”?

 L114 “KITTI dataset.” – “the KITTI dataset.” The same applies to L146.

 L117 “bird eye view” – “bird’s-eye view”

 L130 “Fig. 1 shows ...” – “Figure 1 shows ...”: The general rule is that such an abbreviation should be spelled out at the top of a sentence. (L131 “In Fig. 1, ...” is no problem.)

 L135 “from 16Ch(a) to 64Ch(e)” – “from (a) 16-Ch to (e) 64-Ch.” “(C)bilinear” should be “(c) bilinear.” Please add the axis labels “x”, “y”, and “z” in each panel (or at least the first panel.) It seems two or three different (but similar) colors are employed in dots. If so, please clarify the meaning of each color in the caption. Using more distinguishable color sets would be better if that is the case. The same applies to Fig. 2. [Why are gray dots employed only in Fig. 2(c)?]

 L139 “[5] model resembles ...” – “The simulation-based model [5] resembles ...”

 L141 The term “pipeline” has not been used in the text. Please paraphrase it, or explain the word beforehand.

 L153 An italic font “D” should be used for “and D is ...”

 L159-160 Please rephrase the part “the channel of typically range measurement and intensity.”

 L161 “Research Strategy for Proposed Method” – “Research strategy for the proposed method”

 L163 “given as Eq. (1):” – “given as”: This suggestion is according to the general rule that an equation (or a set of equations) is part of a sentence. Accordingly, it is desirable to put a period just after the third equation.

 L169 “when there is a meaningless zero value” – “when there are meaningless zero values”

 L169 A period is required just after “LiDAR points.”

 L173 (the paragraph) According to the general rule, “6 neighbor pixels” should be “six neighbor pixels.” Similarly, “4-point depth interpolation” – “four-point depth interpolation.” The same applies to L292 and L293.

 L173 “proposed in [31].” – “proposed by Lim et al. [31].”

 L173 “in the counter-clockwise direction” – This statement is not appropriate for the situation depicted in Fig. 5. Maybe “starting from the top-left pixel, as shown in Fig. 5” would be better.

 L173 A comma should be added just after eq. (3). The same applies to eqs. (5) and (6).

 L178 “owing to the interpolation with the neighbor points that has zero or maximum range value” – “owing to the interpolation with the neighbor points that have zero or maximum range values.”

 L179 It is better to move L182-183 just after “to the surrounding data,” namely, “to the surrounding data. These zero or maximum valued pixels in the 2D range image refer to the LiDAR beam points that do not project on the object in the measurable range.”

 L192 “This interpolation result, as shown in Fig. 6(d) clearly removes ...” – “This interpolation result, as shown in Fig. 6(d), clearly removes ...”

 L197-198 The variables, x, y, and z should be in italic. No parentheses are needed for arguments of cos and sin.

 L202 “the ground-truth of 64-Ch.” – “the 64-Ch ground truth.”

 L203 “Fig. 1 and Fig. 2” – “Figs. 1 and 2”

 L215-216 “the mAP(mean average precision) of easy, moderate, and hard difficulty classes of cars, pedestrians, and cyclists, respectively” – This part is misleading since there is no correspondence such as (cars – easy), (pedestrians – moderate), and (cyclists – hard), as indicated in Table 1. A better way would be something like “the mAP (mean average precision) of easy, moderate, and hard difficulty for each class of cars, pedestrians, and cyclists.”

 L220 Fig. 6: The axis labels should be “x/m”, “y/m”, and “z/m”, where only variables (x, y, and z) should be in italic. The same labels should be applied to Figs. 1 and 2 (not all the panels, but the first panel in each case would suffice.)

 L221 “on 2D range image and (D) ...” – “on 2D range image, and (D) ...”; “3d object detection.” – “3D object detection.”

 L226 “using Eq. (2) and 3.” – “using Eqs. (2) and (3).” The same applies to L235.

 L228-230 “The interpolation based on ... enhancement.” – Citing an example (for instance, 51.32% -> 43.85%) would be helpful for the reader to avoid any possible misunderstanding.

 L235 In the header of Table 1, please insert spaces, as “Baseline (32-Ch)” and Ground Truth (64-Ch).”

 L242-243 “The axis of the image was the azimuth and vertical angles, as shown in Fig. 4.” – “The axes of the image are the azimuth and vertical angles, as shown in Fig. 4.” The general rule is that the present tense should be used to explain the Figures and Tables.

 L245 “with the Eq. (1).” – “with Eq. (1).”

 L248-249 “The mAP of easy level cyclist detection was increased to 64.9% from 58.1% of the baseline.” – “The mAP of easy level cyclist detection has been increased from 58.13% of the baseline to 62.45%.”

 L252 “Fig. 7 and Fig. 8” – “Figures 7 and 8”

 L253 “upsampled” – “up-sampled”

 L254 “that helped in” – “that helps in”; “pedestrian and vehicles.” – “pedestrians and vehicles.”

 L264 “The 3D detection result in Table 1 showed that the ground-removal resulted in ...” – “The 3D detection results in Table 1 show that the ground-removal has resulted in ...”

 L267 “Previous methods comparison” – “Comparison with previous methods”

 L271 Fig.7(b) “with 64 ch Point Cloud” – “with 64-Ch point cloud”; the same applies to Fig. 8(b). Fig. 7(c) “with Proposed Upsampling Method” – “with proposed up-sampling method”; the same applies to Fig. 8(c).

 L273 “were compared” – “are compared”

 L274 “The abbreviation of ...” – “The abbreviations of ...”

 L276 “the work by Shan et al. [5]showed ...” – “the work by Shan et al. [5] showed ...”

 L278-279 “they did not reconstruct the object shapes properly to increase the 3D detection performance.” – “they did not reconstruct the object shapes properly.”

 L280 “When the moderate-level class detections were compared as in Table 2, it clearly seen that the proposed solution (Case C) exhibited...” – “When the moderate-level class detections are compared as in Table 2, it is clearly seen that the proposed solution (Case C) exhibits ...”

 L282 “The mAP of overall class was 45.4%, which was ...” – “The mAP of the overall class (level M) is 45.36%, which is ...”

 L286 – 287 “with a more vertical angle resolution.” – “with a better vertical angle resolution.”

 L292 “The interpolation on empty space were ...” – “The interpolation on empty space was ...”

 L297 “the baseline 32-ch point cloud data and then passing them as ...” – “the baseline 32-Ch point cloud data, and then, passing them as ...”

 L298 Table 2: “Comparison Table of Object Detection Performance using Pretrained PointPillars Model (upsampled from 32ch to 64ch).” – “Comparison table of the object detection performance using the pre-trained Point Pillars model (up-sampled from 32 to 64 Ch).” In the row header: “Interp.” – “Interpolation”

Author Response

We have answered to the reviewer comments and revised the manuscript to improve the overall quality.
